# The Level of Fear in the Polish Police Population during the COVID-19 Pandemic with the Impact of Sociodemographic Variables

**DOI:** 10.3390/ijerph19159679

**Published:** 2022-08-05

**Authors:** Barbara Szykuła-Piec, Robert Piec, Artur Zaczyński, Rafał Wójtowicz, Sławomir Butkiewicz, Ewa Rusyan, Kamil Adamczyk, Irena Walecka, Anna Dmochowska, Wioletta Rogula-Kozłowska

**Affiliations:** 1Faculty of Safety Engineering and Civil Protection, The Main School of Fire Service, 01-629 Warsaw, Poland; 2Clinical Department of Neurosurgery, Central Clinical Hospital of the Ministry of the Interior and Administration, 02-507 Warsaw, Poland; 3Anesthesiology and Intensive Care Department, Solec Hospital, 00-382 Warsaw, Poland; 4Department of Conservative Dentistry, Medical University of Warsaw, 02-097 Warsaw, Poland; 5Department of Dermatology, Centre of Postgraduate Medical Education, Central Clinical Hospital of the Ministry of the Interior and Administration, 02-507 Warsaw, Poland

**Keywords:** COVID-19, safety of policemen, mental health, sociodemographic variables

## Abstract

This study has a twofold objective. First, we aim to measure the levels of fear among Polish police officers using the COVID-19 Fear Scale (FCV-19S) that has a stable unidimensional structure allowing for the provision of additional data by combining variables. This structure allows the second objective to be met to measure the correlation with sociodemographic variables. The utilitarian objective of the study is to provide information for updating support policies for stress management in the service. The questionnaire was completed by 1862 people with a mean age of 38.75 years with a good Cronbach’s alpha (0.89). The perceived level of fear associated with COVID-19 should be considered relatively low. Caring for the elderly does not affect the level of fear. The factors of gender, age and having children statistically significantly differentiate the perceptions of fear. Therefore, there is a necessity to focus on building support for police officers who are over 50 years old, as well as for women, where higher levels of fear in both men and women can translate into the development of psychosomatic illnesses.

## 1. Introduction

The SARS-CoV-2 virus that caused COVID-19 has gripped the world causing a pandemic that is proving to be fatal for some vulnerable people and causing a considerable impact on world economies, creating job loss and a considerable amount of suffering related to this. COVID-19, which primarily originated among individuals in Wuhan (China), has spread globally, affecting both physical and mental health. The disease is most commonly characterized by respiratory infections ranging from a common cold to more severe diseases, such as Middle East Respiratory Syndrome (MERS) and Severe Acute Respiratory Syndrome (SARS). It is a major health catastrophe that needs more attention in order to be eradicated. This outbreak has been accompanied by mental stress, fear, depression, insomnia, denial, anger and fear among individuals. Policy makers and health organizations creating prevention strategies are influenced by these collective concerns influencing daily behaviors and the economy, which can weaken control strategies of COVID-19 leading to more morbidity and mental health needs at a global level [1]. In order for decision makers to implement new or update existing mental health support policies, they need to be provided with the relevant data.

A research team consisting of academics from a hospital, which supports mental and physical health police and other services in the area of civilian safety, and the Main School of Fire Services, which conducts research on individual and social resilience, among other topics, decided to collect data on the Polish police.

The answer to the research question concerning the level of fear caused by the SARS-CoV-2 virus, which investigates the correlation between these fear levels and sociodemographic variables in Polish police officers, is the main aim of the present article.

## 2. Background

As police officers are involved in frontline work, they be-came the focus of the present study following the study of health workers. The research team hypothesized that police officers present strong symptoms of fear associated with COVID-19. This is likely to be related to the fact that they play a central role in maintaining public safety, thereby increasing the risk of exposure through interactions with the community.

Police officials work in extremely difficult environments for long hours and carry a significant burden of exposure to COVID-19. Policing is considered as one of the most mentally taxing occupations, competing with continuous threats of violence, rotating shifts, an increased need for hypervigilance and a lack of public support creating chronic stress [2,3,4,5]. As a result, law enforcement officers suffer from mental stress at a higher rate than the general population [2]. Law enforcement officers are easily susceptible to infection as they are not immune from the stress that COVID-19 places among the population. They are considered to be one of the “essential workers” who work and respond to calls for service while others stay safe at home.

Officers have been advised by urban agencies to exercise discretion in conducting traffic stops and avoid unnecessary interpersonal interactions [6]. Overall, calls for service appear to have slightly decreased during the height of the pandemic [7].

Even with a reduction in the number of calls to the police under shelter-in-place guidelines, the burden on individual officers appears to be increasing, as indicated by the fact that COVID-19 killed five times as many police officers as shootings during the US pandemic. COVID-19 has become the leading cause of officer deaths, even though law enforcement agencies were among the first groups eligible to receive the vaccine in late 2020 [8]. In Poland, the police, as in other countries, in addition to performing their statutory tasks, such as detecting offenders, policing and ensuring security, continue to support sanitation services in the fight against COVID-19 in order to reduce its transmission. Every day, since the start of the pandemic, around 20,000 officers have been deployed to combat it. At the end of May 2020, around a hundred police officers were already found to have COVID-19. From the start of operations until the end of 2020, more than 42 million quarantine referral checks were conducted on people, 370,000 checks on public transport, almost 1 million retail outlets, almost 4500 discos and almost 11,500 wedding venues [9]. It can be assumed that such activities increase the risk of infection, which could theoretically translate into a fear of the virus.

In Poland, there are 189 psychologists serving or employed by the police (as of 31 December 2019). Police psychologists work in three areas, or so-called specializations: psychological care and psycho-education, human resource management psychology and police applied psychology.

However, the forms of assistance are mainly traditional and statistics on the mental health of uniformed officers are largely a mystery. Information about thousands of neurotic police officers would probably undermine the prestige of the profession. Undoubtedly, there is a need for building resilience in the form of the capacity to prepare, recover and adapt in the face of stress, challenge, adversity and trauma [10]. Many times, officers are overwhelmed by the scale and quality of traumatic events: traffic accidents with a high number of victims, often women and children; the corpses of newborn babies on rubbish dumps; bodies dredged up from bodies of water or the particular cruelty of perpetrators to victims of violence; responsibility for the weapons entrusted to them; official subordination; working overtime and the danger of losing their life and deteriorating health. Additionally, officers intervene almost daily in difficult environments and are exposed to blood, people suffering from infectious diseases and those under the influence of psychoactive substances.

The mental trauma not resulting from a singular event reinforces occupational stress experienced on a day to day basis due to a prolonged crisis [11].

COVID-19 will impose an everlasting impact on society and police departments, which may end up having to change protocols on many policing strategies on a permanent basis. There will be need to deal with people who have been impacted psychologically, sociologically and economically, and a need to address increased mental health issues in individuals representative of safety services when trying to reestablish public interactions, as more people start coming out.

To fear the unknown, especially if it affects our health and that of our loved ones is a normal reaction. If one cannot determine who has the virus, then the level of fear increases.

## 3. Materials and Methods

Study design: The COVID-19 pandemic has caused colossal disruptions in all areas of the functioning of nations and societies. Law and order is guarded by the police to prevent social unrest and provide support, response and control when required.

The police plays a vital role in a properly functioning state, and in particular in providing an internal security system for the citizens. There is no doubt that, in their daily duties, police officers are exposed to a unique set of challenges, which may increase not only the risk of stress, but may also cause fear, especially during tasks for which they are exposed to direct contact with people who pose a potential threat of being infected with the virus. It is important to distinguish between different types of fear. Fear can occur in the face of a real threat, whereas another form of fear is irrational, arising from either an imagined danger or anticipated threat. Fear is the most common psychopathological symptom.

Fear causes the secretion of substances from our body that puts us in a state of readiness to flee or fight. The adrenaline produced in a stressful situation induces, for example, an accelerated heartbeat, an increase in blood pressure and the secretion of sweat. Therefore, we decided to check to what extent this professional group experienced the fear of becoming infected by COVID-19.

In this study, use was made of a tool developed by a team of researchers from the University of Hong Kong (Ahorsu et al.): the COVID-19 Fear Scale (FCV-19S) [12]. The English version of the scale questionnaire was translated into Polish by two independent translators. The resulting texts were then compared, merged and analyzed to best reflect the original version. The FCV-19S scale was validated for use as a reliable research tool in many studies [13,14,15,16,17,18].

All the obtained results were statistically analyzed with the use of PQStat for Windows Software and SPSS v 21. The entire statistical analysis was performed at the confidence level α = 0.05. The results for which the probability of making a type-1 error consisting in the rejection of the null (true) hypothesis was less than 0.05 were considered significant.

In this study, the internal consistency of the scales was verified by using Lee Cronbach’s alpha coefficient and determining the correlation coefficients between the answers to the individual questions and the total scale score. The minimum level of Lee Cronbach’s alpha was assumed to be 0.7 (Table 1, Figure 1).

A cross-sectional study was conducted between 1 and 30 June 2020 using a survey questionnaire (paper version) to be completed in person. During this period, 46,548 people died from COVID-19, ranging from 22,000 to 30,000 active cases, with an average of 400 cases per day (data from the Health Ministry (https://basiw.mz.gov.pl/ (accessed on 30 July 2020))).

The trial consisted of 1885 active police officers serving in the Mazovia Province. The study was approved by the Bioethics Committee of the Warsaw Medical University under the number AKBE/143/2020. Written informed consent was obtained from all participants.

An analysis of the data obtained from the questionnaire was performed, which consisted of demographic questions to characterize the study sample and questions to determine the fear levels according to the FCV-19S. The questions were of a closed nature. The level of fear was assessed on a 5-point scale, where 1 meant that the respondent did not have the given symptom and 5 meant that the respondent felt it strongly.

Patients’ fear of COVID-19 was measured by the following questions and statements:I am afraid of the coronavirus (COVID-19) more than anything else;Thinking about the coronavirus (COVID-19) makes me uncomfortable;My palms sweat when I think about the coronavirus (COVID-19);I am worried that I will die from the coronavirus infection (COVID-19);When I hear about the coronavirus (COVID-19) in the media and on the Internet, I become nervous and worried;I cannot sleep because of the coronavirus (COVID-19);My heart starts to beat faster when I think about the coronavirus (COVID-19).

New variables were created: factor 1: emotional (mental) fear reactions (questions 1, 2, 4 and 5), factor 2: symptomatic expressions of fear (questions 3, 6 and 7) and factor 3: the general level of fear, consisting of all 7 symptoms of fear (questions 1–7). The flow diagram of survey show on the Figure 2.

The questionnaire was voluntary and anonymous, and was distributed to police officers working in the Mazovia Province who were being tested for COVID-19 at the Hospital of the Ministry of the Interior and Administration in Warsaw on the day of the survey.

Study size:

There were 96,132 police officers, including 61,789 (64.28%) men and 34,343 (35.72%) women [19]. The questionnaire was completed by 2008 officers and 1885 questionnaires were qualified for further analysis after assessing the correctness of the data entered, including 34.11% women and 65.89% men. In the surveyed group, the proportions of men and women were very similar to the general gender distribution among police officers in Poland. Only questionnaires in which the age of the respondents ranged from 20 to 66 years and all answers in the relevant part were marked were admitted for further analysis.

For the entire population of police officers, assuming a significance level and margin of error of 0.05, the required number of persons undertaking the survey was 382.6.

The margin of error for the collected data was calculated:Confidence level (α): 95%;Sample size (n): 1887;Proportion percentage (p): 50;Population size (N): 96,132.

The error margin was equal to ±2.235% [20].

Responses were coded using a five-point Likert scale as follows: 1 for “definitely not” to 5 for “definitely yes”, with the option to mark “3” if the opinion was imprecise. In the new variables constructed on the basis of the answers obtained, it was established that if the mean was less than or equal to 2.5, the respondent was not afraid; the variable took the value of 0 if the mean was equal to or greater than 2.51 and if the respondent was afraid, the variable took the value of 1. SPSS Statistics version 21.0 and PQStat for Windows software was used to perform statistical analyses, and the chi-squared test and Pearson’s correlation coefficient were applied. Logistic regression analysis was performed for the newly created variables.

## 4. Results

A total of 2008 people filled in the online questionnaire and 1885 questionnaires were then qualified for further analysis after evaluating the correctness of the data entered, including 34.11% of women and 65.89% of men. The questionnaire was completed by police officers aged between 20 and 66 years (the average age was 39 years). Most of the respondents, 72%, had children. Almost every fifth police officer lived with or took care of elderly people. Detailed data are presented in Table 2.

The surveyed group of police officers indicated the highest level of fear (mean 2.42) when stating (Table 3): I fear the coronavirus (COVID-19) more than anything else. The highest symptomatic fear symptoms were recorded for the question: MY PALMS SWEAT WHEN I THINK ABOUT the CORONAVIRUS (mean 1.47) and MY HEART STARTS TO BEAT FASTER WHEN I THINK ABOUT the CORONAVIRUS (mean 1.46); for both symptoms, about 2% indicated levels 4 and 5 (high levels of fear). The level of emotional symptoms of the drug (questions 1, 2, 4 and 5; averages 2.13) was significantly higher than the physical symptoms (questions 3, 6 and 7; averages 1.44).

For 78% of the study population, COVID-19 was not a cause of fear (marked values 1 and 2). Almost every 10th person (7.87%) marked 4 or 5 for the individual questions indicating high fear levels towards the coronavirus. The low standard deviation indicates a clustering of results close to the mean.

It was then investigated whether gender, age, having children and caring for the elderly affected fear levels. For this purpose, a chi-squared test was performed for the nominal variables, and the Pearson linear correlation coefficient of the variable age was calculated.

The performed analysis indicates that for a=0.05, gender significantly differentiates the way that fear is perceived. It can also be observed that the older the respondent, the higher the level of fear he/she feels. Having children also significantly affects the level of fear; only in the case of sweating hands can no statistically significant difference be assumed. Taking care of the elderly, on the other hand, does not affect the feeling of fear (Table 4).

A two-factor model of the COVID-19 fear measure was then developed. It was proposed to create new variables: mental (emotional) fear reactions (questions 1, 2, 4 and 5) and somatic fear symptoms (questions 3, 6 and 7). If the mean value of responses was up to 2.5, it was considered that the respondent was not afraid and the variable took the value of 0; a mean value higher than 2.50 meant that the respondent was afraid and the variable took the value of 1. Additionally, a model generalizing all symptoms was created—the general level of fear, consisting of all seven symptoms of fear (questions 1–7); an analogous distribution depending on the mean was adopted. After constructing the new variables, logistic regression analysis was performed.

The results point to the following (Table 5, Figure 3):Gender: OR [95%] = 0.49 [0.393942; 0.602288], *p* = 0.000, the odds of mental (emotional) symptoms in a woman are 0.5 fold higher than in a man;Age: OR [95%] = 1.07 [1.052602; 1.079336], *p* = 0.000, the odds of mental (emotional) symptoms are slightly higher in older people;Having children: OR [95%] = 1.83 [1.419435; 2.346546], *p* = 0.000, the odds of mental (emotional) symptoms in a person with children is 1.83 fold higher than in people without children;Caring for the elderly is not statistically significant.

The results point to the following (Table 6, Figure 4):Gender: OR [95%] = 0.68 [0.481045; 0.950956], *p* = 0.024, the odds of somatic symptoms in a woman are higher than in a man;Age: OR [95%] = 1.05 [1.026699; 1.066078], *p* = 0.000, the odds of somatic symptoms are slightly higher in older people;Having children: OR [95%] = 1.51 [1.004557; 2.277944], *p* = 0.048, the odds of somatic symptoms in a person with children is 1.53 fold higher than in those without children;Caring for the elderly is not statistically significant.

The results point to the following (Table 7, Figure 5)):Gender: OR [95%] = 0.60 [0.469761; 0.770796], *p* = 0.000, the odds of developing symptoms are slightly higher in older people;Age: OR [95%] = 1.06 [1.046349; 1.076548], *p* = 0.000, the odds of symptoms are slightly higher in older people;Having children: OR [95%] = 1.73 [1.280751; 2.333964], *p* = 0.000, the odds of having emotional symptoms in a person with children is 1.73 fold higher than in people without children;Caring for the elderly is not statistically significant.

In the next step of the analysis, the age variable was divided into three groups: 1: from 20 to 35 years of age, 2: from 36 to 50 years of age, 3: above 50 years of age. A logistic regression model was calculated for individual and total symptoms, respectively.

The likelihood of psychological (emotional) effects of fear in people older than 50 years increased 2.65 times (Table 8 and Table 9).

The odds of somatic effects of fear in people aged older than 50 years increased 2.39 fold (Table 10 and Table 11).

The odds of experiencing the effects of fear overall increased by 2.2 fold in people aged older than 50 years. The prediction for different variants of the variables was also used to analyze the model (Table 12 and Table 13).

The results show that the likelihood of fear symptoms increases for women, people with children and those aged over 50 years (Table 14).

## 5. Discussion

The COVID-19 pandemic undeniably resulted in a stressful time for the whole world, with people facing daily concerns about their own health and the lives and those of their loved ones, as well as fears associated with an uncertain economic situation. This stress especially concerned the services that, irrespective of the epidemic situation, guarded the order and safety of citizens. Those who decided to work in the police force, consciously took the risk of serving on the front line in the face of various threats. They were the people who, in dangerous situations, would stay at the police station. The current COVID-19 pandemic is just such an emergency, and one that has lasted for over a year. Regardless of the number of confirmed cases of the coronavirus, which is increasing by the day, police officers must perform their duties. On a daily basis, Polish police officers intervene more than 15,000 times, which means that this is how many times they are potentially exposed to COVID-19. American Resilience expert Mike Taigman together with Sascha Liebowitz composed the book “Super-Charge Your Stress Management in the Age of COVID-19”, which was written after the authors gathered information on the intense suffering and stress associated with COVID-19 among emergency services and healthcare providers. In the book, they compiled a number of techniques for coping with stress in this particular situation that are easy to learn and apply in any situation [21]. In turn, the research conducted by Dr Michelle Lilly (Associate Professor of Clinical Psychology at Northern Illinois University) and Sergeant Shawn Curry (an active-duty officer with the Peoria Police Department in Peoria, Illinois) was concerned with the main question: What is the state of officers’ mental health in 2020? Does this indicate high levels of stress (distress) among police officers [22]? The study was conducted in August 2020. Over a two-week period, a total of 1355 officers were interviewed in an online survey. Almost half of the respondents (47%) had a result indicating that they might develop PTSD (post-traumatic stress disorder), 29% declared moderate to very severe fear levels, while 37% declared moderate or very severe depression. When analyzing these results, it is important to bear in mind the circumstances, time and environment of the research. It is difficult to make a clear comparison, if only considering the types of tools used along with the environmental and cultural differences between Poland and the United States of America; nevertheless, the results of our study demonstrate a lower level of fear during the current pandemic. Most current research focuses on medical personnel and university students, and no attention has been paid to police officers [15,23,24,25]. Of course, over time, further studies and articles (among others, such as [26,27])) describe this problem.

In our study, the highest level of fear was recorded for the statement that the coronavirus (COVID-19) was feared by respondents more than anything else, with 16.1% of respondents reporting high and very high levels of fear and 25.8% reporting medium levels. It can be assumed that this was dictated by the relatively reduced knowledge of the virus, its actual spread, its power of destruction and its impact on the human body. In short, we are afraid of what we do not know. One quarter of respondents found it difficult to define their fear of the coronavirus. Physical symptoms of fear were not noted, which bodes well for the potential development of PTSD; nevertheless, it is vital to keep observing one’s own behavior and the behavior of colleagues in order not to overlook any alarming symptoms, because, as we know, they may manifest themselves only after some time (even up to approximately half a year). The analysis of the collected data indicated that gender, having children and age statistically significantly differentiated feelings of fear [28,29]. The studies conducted, to date, indicate inter-individual differences in stress reactivity among police officers, according to gender and work experience [30].

A review of the accumulated data showed that gender, having children and age significantly differentiated feelings of fear. Taking care of the elderly, on the other hand, did not affect the feeling of fear. The probability of occurrence of both mental and somatic symptoms in a woman was about 0.5-fold higher than in a man. The probability of experiencing fear symptoms by a person with children was almost twice as high as that of a person without children. The likelihood of the effects of fear in people aged over 50 years increased more than two fold. A predictive analysis for different variants of variables confirmed the previous analyses; the probability of fear symptoms occurring increased for women, people with children and those aged over 50 years. The questionnaire was completed only by persons working in the Mazowieckie Province, but it is the largest of all provinces in Poland, with towns of various sizes and small villages. Taking into account the sample size, the level of significance, margin of error and gender and age distribution of the respondents, the authors are inclined to believe that the results can be generalized to the entire population of police officers in Poland. This tool can be used to raapidly screen the level of fear among uniformed services and minimize its serious negative consequences.

## 6. Conclusions

The COVID-19 pandemic has affected all spheres of our lives. It particularly affected the emergency and order services, which are on active duty at all times. Therefore, it is reasonable to define the effects that the current situation may bring in order to strengthen the resilience of people and systems to this crisis. Despite the relatively low level of fear recorded, it is still necessary to provide resources that can be reached in the event of a mental crisis. The resources come in the form of correct, functional relationships, acquiring knowledge and skills in the techniques of coping with stress after its prior self-diagnosis. This is all the more relevant as the current situation in which we constantly use our internal resources to deal with stress is difficult to determine.

Therefore, it is important, in addition to monitoring the impact of COVID-19 on the psychophysical wellbeing of officers, to prepare response plans, procedures, training and logistic facilities in the form of personal protective equipment or disinfectants.

Generally, the surveyed group of police officers indicated the highest level of fear when stating: I fear the coronavirus (COVID-19) more than anything else. The level of mental symptoms of the drug was significantly higher than the physical symptoms. Fear is a form of subjective, mental perception of a phenomenon. Therefore, acquiring information, knowledge and strengthening mental resilience can be a recommended form of reducing fear levels.

The performed analysis indicated that gender, age and having children significantly differentiated the way that fear was perceived. The older the respondent, the higher the level of fear he/she feels. Taking care of the elderly, on the other hand, does not affect the feeling of fear.

Investigating the relationship between experiencing fear and age, the gender and family situation makes it necessary to pay attention to building support for police officers over 50 years of age and also for women, in whom higher levels of fear, related to the structure of the brain [31], combined with responsibility may translate into the development of psychosomatic diseases. Above all, a pandemic is fundamentally different from previous emergencies, and prevention and control efforts are arduous. The pandemic will have a lasting impact on society worldwide for a long time. During the pandemic, police officers were the main force used for maintaining social stability and, consequently, their mental health was a determinant for success or failure, at least to some extent, of efforts to prevent and control the pandemic, but also to maintain public safety. The field of law enforcement faces the most challenges considering mental and emotional wellbeing, even in the absence of a global pandemic. It is well known that police officers are exposed to uncontrolled stress and suffer from high rates of fear and depression. Over 20,000 law enforcement agencies across the nation with more than 750,000 individuals felt this type of stress and resulting PTSD during this pandemic [32]. The above-mentioned health issues will have a direct impact on the officers, both professionally and personally, which is evidenced by increased burnout rates, job turnovers and increased cases of alcohol abuse, divorce and suicide in comparison to the general public outside the profession. One of the reasons officers live shorter lives compared to the general public can be attributed to stress, which is directly linked to heart diseases [33]. The research also shows that police officers have high rates of PTSD, which is credited with a range of health outcomes, including suicide.

The fact that more officers will die as a direct result of contracting the virus in the course of serving their communities is itself a reminder of the sacrifices these officers make. Sadly, the cost of this pandemic will not end with just life loss for the law enforcement community. Based on the historical accounts of similar illnesses, such as SARS, the indications point to other long-lasting complications for the survivors of this pandemic. The COVID-19 pandemic is not abating and, as a result, frontline emergency services, whether the police, fire service or emergency ambulance service, are experiencing chronic strain. Based on the data obtained from the English charity organization Mind, which focuses on mental health, it was claimed that almost 90% of emergency service staff have experienced stress, bad moods and poor mental health at some point during their work for the emergency services. To date, as Professor Wankhade pointed out, there has been an increase in cases of post-traumatic stress disorder (PTSD) among rescuers [34]. This, in turn, undoubtedly affects mental wellbeing and, above all, the capacity to function effectively at work.

In the months that follow, it is important that government officials, decision makers and scholars use COVID-19 as an opportunity to collate experiential knowledge and insights from those working on the frontline.

## Figures and Tables

**Figure 1 ijerph-19-09679-f001:**
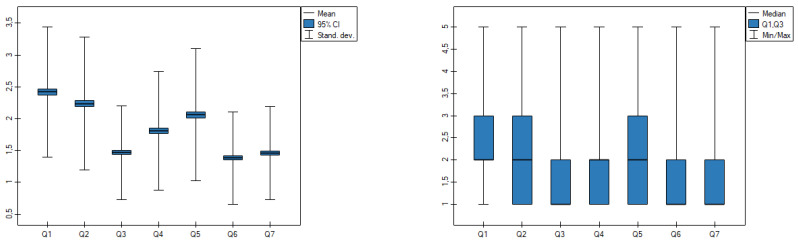
Mean and median level of fear. Source: own study based on research.

**Figure 2 ijerph-19-09679-f002:**
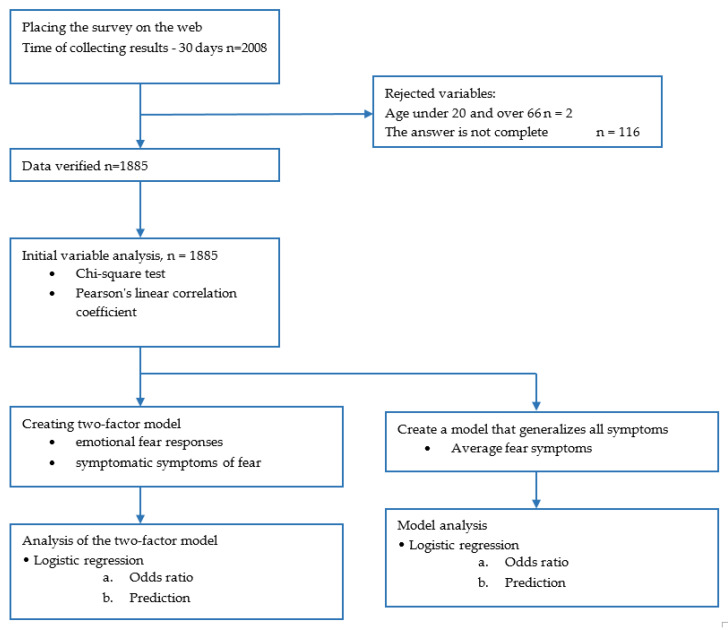
Flow diagram. Source: own study.

**Figure 3 ijerph-19-09679-f003:**
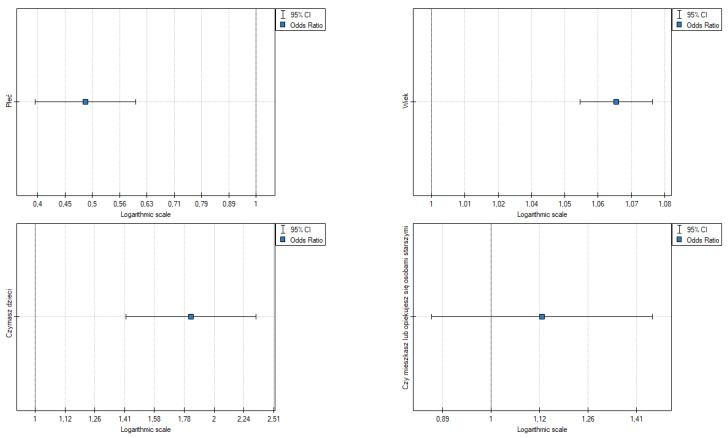
Odds ratios for psychological (mental) symptoms of fear and demographics. Source: own study based on research.

**Figure 4 ijerph-19-09679-f004:**
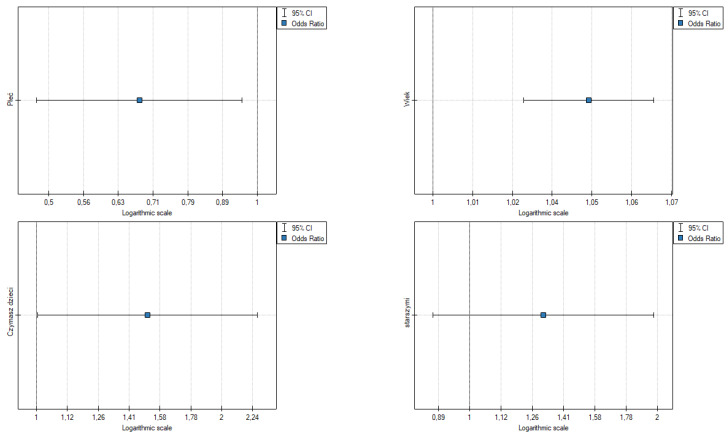
Odds ratios of somatic symptoms of fear and demographics. Source: own study based on research.

**Figure 5 ijerph-19-09679-f005:**
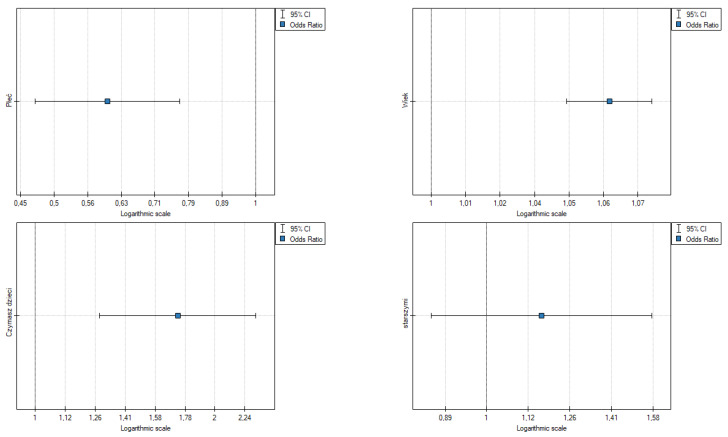
Odds ratios of total fear symptoms and demographic data. Source: own study based on research.

**Table 1 ijerph-19-09679-t001:** Cronbach’s alpha test.

Cronbach’s Alpha/Split-Half	
Analyzed variables	Q1; Q2; Q3; Q4; Q5; Q6; Q7
Significance level	0.05
Group size	1885
Number of items	7
Mean of scale	12.84191
Standard deviation of scale	4.872629
Cronbach’s alpha for scale	0.88701
−95% CI for Cronbach’s alpha for scale	0.879043
+95% CI for Cronbach’s alpha for scale	0.894633
Standard error of measurement	1.637888
Average correlation between pairs of items	0.539206
Standardized Cronbach’s alpha	0.8912

Source: own study based on research.

**Table 2 ijerph-19-09679-t002:** Demographic information.

Demographics	Group	N (%)/Min, Max, Mean
Gender	Woman Man	643 (34.1%) 1242 (65.9%)
Age (in years)		Min 20, max 66, mean 38.75
Do you have children?	No	528 (28%)
	Yes	1357 (72%)
Do you live with any elderly persons or are you taking care of them?	No	1540 (81.7%)
	Yes	345 (18.3%)

Source: own study based on research.

**Table 3 ijerph-19-09679-t003:** Distribution of answers to individual questions.

Fear Symptoms	Group	Frequency *n*	Percent (%)	Mean	Std. Deviation
I am afraid of the coronavirus (COVID-19) more than anything else	1	350	18.6	2.42	1.02
	2	745	39.5		
	3	486	25.8		
	4	256	13.6		
	5	48	2.5		
Thinking about the coronavirus (COVID-19) makes me uncomfortable	1	511	27.1	2.24	1.05
	2	722	38.3		
	3	384	20.4		
	4	229	12.1		
	5	39	2.1		
My palms sweat when I think about the coronavirus (COVID-19)	1	1228	65.1	1.47	0.74
	2	475	25.2		
	3	145	7.7		
	4	30	1.6		
	5	7	0.4		
I am worried that I will die from the coronavirus (COVID-19) infection	1	851	45.1	1.81	0.93
	2	682	36.2		
	3	238	12.6		
	4	82	4.4		
	5	32	1.7		
When I hear about the coronavirus (COVID-19) in the media and on the internet, I become nervous and worried	1	645	34.2	2.06	1.04
	2	755	40.1		
	3	250	13.3		
	4	193	10.2		
	5	42	2.2		
I cannot sleep because of the coronavirus (COVID-19)	1	1366	72.5	1.38	0.73
	2	370	19.6		
	3	111	5.9		
	4	21	1.1		
	5	17	0.9		
My heart starts to beat faster when I think about the coronavirus (COVID-19)	1	1237	65.6	1.46	0.74
	2	484	25.7		
	3	121	6.4		
	4	36	1.9		
	5	7	0.3		

Source: own study based on research.

**Table 4 ijerph-19-09679-t004:** Statistical significance of the chi-squared test and Pearson correlation coefficient values.

Fear Symptoms	Gender	AGE	Having Children	Taking Care of the Elderly
I am afraid of the coronavirus (COVID-19) more than anything else	0.00	Pearson Correlation R = 0.28 Sig. = 0.00	0.00	0.018
Thinking about the coronavirus (COVID-19) makes me uncomfortable	0.00	Pearson Correlation R = 0.27 Sig. = 0.00	0.00	0.261
My palms sweat when I think about the coronavirus (COVID-19)	0.009	Pearson Correlation R = 0.10 Sig. = 0.00	0.114	0.614
I am worried that I will die from the coronavirus infection (COVID-19)	0.00	Pearson Correlation R = 0.23 Sig. = 0.00	0.00	0.692
When I hear about the coronavirus (COVID-19) in the media and on the Internet, I become nervous and worried	0.00	Pearson Correlation R = 0.27 Sig. = 0.00	0.00	0.226
I cannot sleep because of the coronavirus (COVID-19)	0.022	Pearson Correlation R = 0.15 Sig. = 0.00	0.00	0.726
My heart starts to beat faster when I think about the coronavirus (COVID-19)	0.004	Pearson Correlation R = 0.17 Sig. = 0.00	0.003	0.099

Source: own study based on research.

**Table 5 ijerph-19-09679-t005:** Logistic regression model for psychological (mental) fear symptoms and demographics.

	b Coeff.	b Error	−95% CI	+95% CI	Wald Stat.	*p*-Value	Odds Ratio	−95% CI	+95% CI
**intercept**	0.099904	0.179553	−0.252014	0.451822	0.309586	0.577934	1.105065	0.777234	1.571173
**Gender**	−0.719285	0.108301	−0.931551	−0.50702	44.110373	<0.000001	0.4871	0.393942	0.602288
**intercept**	−3.609458	0.267063	−4.132891	−3.086024	182.665665	<0.000001	0.027067	0.016036	0.045683
**Age**	0.063806	0.006398	0.051265	0.076346	99.450684	<0.000001	1.065885	1.052602	1.079336
**intercept**	−2.118462	0.234419	−2.577916	−1.659009	81.668484	<0.000001	0.120216	0.075932	0.190328
**Do you have children**	0.601602	0.128238	0.350259	0.852944	22.00805	0.000003	1.825039	1.419435	2.346546
**intercept**	−1.208915	0.168249	−1.538677	−0.879152	51.627992	<0.000001	0.298521	0.214665	0.415135
**Do you take care of the elderly**	0.120665	0.134059	−0.142084	0.383415	0.810172	0.36807	1.128247	0.867548	1.467287

Source: own study based on research.

**Table 6 ijerph-19-09679-t006:** Logistic regression model for somatic fear symptoms and demographics.

	b Coeff.	b Error	−95% CI	+95% CI	Wald Stat.	*p*-Value	Odds Ratio	−95% CI	+95% CI
**intercept**	−1.828853	0.288128	−2.393573	−1.264132	40.289059	<0.000001	0.160598	0.091303	0.282484
**Gender**	−0.391041	0.173857	−0.731794	−0.050288	5.058962	0.024499	0.676352	0.481045	0.950956
**intercept**	−4.2796	0.409519	−5.082243	−3.476957	109.208735	<0.000001	0.013848	0.006206	0.030901
**Age**	0.045168	0.009602	0.026349	0.063987	22.129007	0.000003	1.046203	1.026699	1.066078
**intercept**	−3.188513	0.382667	−3.938526	−2.438499	69.427962	<0.000001	0.041233	0.019477	0.087292
**Do you have children**	0.41391	0.208863	0.004547	0.823273	3.927266	0.047509	1.512721	1.004557	2.277944
**intercept**	−2.787494	0.266642	−3.310103	−2.264886	109.287868	<0.000001	0.061575	0.036512	0.103842
**Taking care of the elderly**	0.270499	0.207137	−0.135481	0.67648	1.705372	0.191587	1.310619	0.873296	1.966941

Source: own study based on research.

**Table 7 ijerph-19-09679-t007:** Logistic regression model for total fear symptoms and demographics.

	b Coeff.	b Error	−95% CI	+95% CI	Wald Stat.	*p*-Value	Odds Ratio	−95% CI	+95% CI
**intercept**	−0.798624	0.20925	−1.208747	−0.388501	14.566407	0.000135	0.449948	0.298571	0.678073
**Gender**	−0.507932	0.126329	−0.755532	−0.260332	16.166097	0.000058	0.601739	0.469761	0.770796
**intercept**	−4.017329	0.308811	−4.622588	−3.412071	169.234628	<0.000001	0.018001	0.009827	0.032973
**Age**	0.059533	0.007258	0.045307	0.073759	67.272773	<0.000001	1.061341	1.046349	1.076548
**intercept**	−2.582963	0.281145	−3.133996	−2.031929	84.406639	<0.000001	0.07555	0.043543	0.131082
**Do you have children**	0.547507	0.153095	0.247447	0.847568	12.789624	0.000349	1.728938	1.280751	2.333964
**intercept**	−1.802264	0.196581	−2.187555	−1.416972	84.052986	<0.000001	0.164925	0.112191	0.242447
**Caring over the elderly**	0.151746	0.155645	−0.153313	0.456804	0.950521	0.329587	1.163864	0.857861	1.57902

Source: own study based on research.

**Table 8 ijerph-19-09679-t008:** Logistic regression model for psychological (emotional) symptoms of fear and three age groups.

Mental	b Coeff.	b Error	−95% CI	+95% CI	Wald Stat.	*p*-Value	Odds Ratio	−95% CI	+95% CI
**intercept**	−0.920283	0.068274	−1.054098	−0.786468	181.689736	<0.000001	0.398406	0.348507	0.455451
**Age of group [1]**	−0.749672	0.124751	−0.994179	−0.505165	36.112298	<0.000001	0.472522	0.370027	0.603406
**Age of group [3]**	0.975853	0.180168	0.622729	1.328976	29.336758	<0.000001	2.653428	1.864009	3.777173

Source: own study based on research.

**Table 9 ijerph-19-09679-t009:** Odds ratios of psychological (emotional) symptoms of fear and age groups.

	OR [95%]	Value of *p*
20 to 35 years old	**0.472522 [0.370027 0.603406]**	**<0.000001**
36 to 50 years old	**Reference**	
Older than 50 years	**2.653428 [1.864009 3.777173]**	**<0.000001**

Source: own study based on research.

**Table 10 ijerph-19-09679-t010:** Logistic regression model for somatic symptoms of fear and three age groups.

Somatic	b Coeff.	b Error	−95% CI	+95% CI	Wald Stat.	*p*-Value	Odds Ratio	−95% CI	+95% CI
**intercept**	−2.432581	0.113127	−2.654306	−2.210856	462.381609	<0.000001	0.08781	0.070348	0.109607
**Age of group [1]**	−0.406805	0.201623	−0.80198	−0.011631	4.070906	0.043628	0.665774	0.44844	0.988437
**Age of group [3]**	0.872333	0.247389	0.38746	1.357206	12.433844	0.000422	2.392487	1.473235	3.885322

Source: own study based on research.

**Table 11 ijerph-19-09679-t011:** Odds ratios of somatic symptoms of fear and age groups.

	OR [95%]	Value of *p*
20 to 35 years	**0.665774 [0.44844 0.988437]**	**0.043628**
36 to 50 years	**Reference**	
more than 50 years	**2.392487 [1.473235 3.885322]**	**0.000422**

Source: own study based on research.

**Table 12 ijerph-19-09679-t012:** Logistic regression model for total fear symptoms and three age groups.

Total	b Coeff.	b Error	−95% CI	+95% CI	Wald Stat.	*p*-Value	Odds Ratio	−95% CI	+95% CI
**intercept**	−1.481605	0.079333	−1.637094	−1.326115	348.785239	<0.000001	0.227273	0.194545	0.265507
**Age of group [1]**	−0.728607	0.150375	−1.023336	−0.433879	23.476787	0.000001	0.482581	0.359394	0.647991
**Age of group [3]**	0.788457	0.193762	0.408691	1.168224	16.558434	0.000047	2.2	1.504846	3.216275

Source: own study based on research.

**Table 13 ijerph-19-09679-t013:** Odds ratios of total fear symptoms and age groups.

	OR [95%]	Value of *p*
20 to 35 years old	**0.482581 [0.359394 0.647991]**	**0.000001**
36 to 50 years old	**Reference**	
Older than 50 years	**2.2 [1.504846 3.216275]**	**0.000047**

Source: own study based on research.

**Table 14 ijerph-19-09679-t014:** Prediction results of selected models for different variants of variables.

Sex	Woman	Man	Woman	Woman	Woman
**Age**	51	51	51	50	55
**Having children**	Yes	Yes	No	Yes	Yes
**Cut-off line**	0.5	0.5	0.5	0.5	0.5
**Probability prediction**	0.50625	0.348061	0.494359	0.491134	0.566323
**Prediction Y**	1	0	0	0	1

Source: own study based on research.

## Data Availability

Not applicable.

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
