# Peer review of "The Level of Fear in the Polish Police Population during the COVID-19 Pandemic with the Impact of Sociodemographic Variables"

_ijerph, 2022, doi:10.3390/ijerph19159679_

Round 1

Reviewer 1 Report

The COVID - 19 pandemic brought numerous qualitative and quantitative changes, positive and negative consequences in all spheres of human life, but also in numerous professions and occupations. Therefore, any research into these changes provides a significant contribution to understanding the pandemic and responding adequately to its impact and consequences. Also, the research and its results enable a proactive response to potential challenges in the sphere of safety and health.

Police officers, in addition to health workers, are on the front line of defense and response to these challenges. Therefore, this research also has a significant scientific and social contribution.

However, a detailed insight into the structure of the work/research, into its methodological and hypothetical approach, points to certain questions and ambiguities that the authors should clarify:

1. In the Introduction, the authors did not explain clearly and convincingly why they chose the police as the target group in the research.

In section 1.1. one should first list previous researches and their findings and only then present the specifics of the police as a service, institution, that is, a part of society made up of professionals who are, above all, ordinary citizens of a specific country.

In section 1.2. the importance of "police training" for specific research is not clearly indicated. The connection of that "training" with the objectives of the research was not explained. I am of the opinion that instead of the term "police training" it would be more appropriate to use the term "psychological support".

Section 1.3. according to its content, it corresponds more to the Introduction, that is, to the introductory considerations.

The analyzed research does not contain a clearly defined hypothetical framework. It should be based on the facts presented in the Introduction, ie. in sections 1.1 to 1.3.

2. In the methodological framework of the research, it is not clearly argued whether fear among police officers is investigated, or stress or anxiety, although in the introductory presentation it was stated that the aim of the research is to examine the relationship between the level of fear and sociodemographic variables among police officers.

The structure of the sample should contain more characteristics of the local police service, that is, the sociodemographic specificities of police officers. Also, it is important to objectively present the contextual framework of the research in order to understand the structure of the sample and the findings of the research itself.

Because of all this, it is not clear whether the research examines stress among police officers and its determinants, i.e., COVID-19 as one of the determinants, or whether it investigates solely COVID-19 as a cause of stress.

3. Within the Discussion, some results of previous research were presented, which should primarily be used as an introduction to the research.

Author Response

Thank You very much for very important and relevant comments.

1. In the Introduction, the authors did not explain clearly and convincingly why they chose the police as the target group in the research - weI hope that the amendments made reflect the clarification

2. We change the section 1.1. , 1.2. and 1.3 acording Your sugestion 

2. We keep the study of fear not anxiety. 
And stated that the aim of the research is to examine the relationship between the level of fear and sociodemographic variables among police officers.

The research examines investigates solely COVID-19 as a cause of fear.

Reviewer 2 Report

As attachment.

Author Response

Thank You for your important remarks.

- Abstract - we briefly explained and modifed

- Introduction - we changed
- The reference number, [35], now is correct
- In the introduction, you emphasized that the COVID-19 epidemic has caused 
greater pressure and negative effects on the work of police. However, the front-line police officers have to face various risks, and their work pressure is very high. Therefore, you have to address the specific impact of the COVID-19 outbreak on police stress, such as the adequacy of protective equipment, pressure of being exposed to more people at risk of contracting the virus, and many evidence or studies about the stress of being concerned about the risk of infecting your family without knowing you have the virus. - yes, but our study did apply only mental and phisical fear. 
Materials and Methods: 
1. Whether the questionnaire was implemented with informed consent? yes - The study was approved by the Bioethics Committee of the Warsaw Medical University under the number AKBE/143/2020. Written informed consent was obtained from all participants.
2. Please explain whether the method of sampling, such as purposive or random 
sampling, can represent the population - yes, it can
3. It should be stated how to distribute and administer the questionnaire, such as in person or entrusted to others - in person, during their visit in hospital for first vacinat
4. It is recommended to add analytical strategies to this unit, so that readers can 
clearly know how you achieved these results - we show our analytical strategies on flow diagram
- Discussion - we modifed

Reviewer 3 Report

The title of manuscript is very broad and do not reveal the main idea, the main aim of the manuscript. The same problem is with the theoretical part and the Discussion: it is difficult to understand to what main question would like to answer the authors with their study, for example, it is not clear why the chapter 1.2 about the training in theoretical part is needed if the aim of the study as write the authors is "to investigate the correlation between the fear-level of COVID-19 and sociodemographic variables in Polish policemen". So, the authors should revise the idea of the study and rewrite the theoretical part and Discussion part, to correct the title.

References to literature sources are missing in the text, for example, line 174.

It seems that the authors use different words fear and anxiety for the same concept, for the same scale. As researchers know the concepts fear and anxiety have different meaning and cannot be used as synonyms.

The conclusions must more accurately reflect the results obtained.

Author Response

Thank You for your very important and valuable comments.
1. The title of manuscript is very broad and do not reveal the main idea, the main aim of the manuscript - we changed

2. The same problem is with the theoretical and the last part (discussion) - we have made a change

3. It seems that the authors use different words fear and anxiety for the same concept, for the same scale. As researchers know the concepts fear and anxiety have different meaning and cannot be used as synonyms - we have, of course, standardised the nomenclature - it was problem during translate

4. The conclusions must more accurately reflect the results obtained - we have made a change

Round 2

Reviewer 3 Report

Thank you for the revision of the manuscript.